# Patient factors affecting successful linkage to treatment in a cervical cancer prevention program in Kenya: A prospective cohort study

Charlotte M. Page [1]*, Saduma Ibrahim[2], Lawrence P. Park[3,4], Megan J. Huchko[1,4]

**1** Department of Obstetrics and Gynecology, Duke University, Durham, North Carolina, United States of America, **2** Kenya Medical Research Institute, Nairobi, Kenya, **3** Department of Medicine, Division of Infectious Diseases, Duke University, Durham, North Carolina, United States of America, **4** Duke Global Health Institute, Duke University, Durham, North Carolina, United States of America

* charlotte.page@duke.edu

**Data Availability Statement:** All relevant data are within the manuscript and its Supporting Information files.

## Abstract

### Objective

To identify patient factors associated with whether women who screened positive for high-risk human papillomavirus (hrHPV) successfully accessed treatment in a cervical cancer prevention program in Kenya.

### Methods

A prospective cohort study was conducted as part of a trial of implementation strategies for hrHPV-based cervical cancer screening in western Kenya from January 2018 to February 2019. In this larger trial, women underwent hrHPV testing during community health campaigns (CHCs), and hrHPV+ women were referred to government facilities for cryotherapy. For this analysis, we looked at rates of and predictors of presenting for treatment and presenting within 30 days of receiving positive hrHPV results ("timely" presentation). Data came from questionnaires completed at the time of screening and treatment. Multivariable logistic regression was used to identify factors associated with each outcome.

### Results

Of the 505 hrHPV+ women, 266 (53%) presented for treatment. Cryotherapy was performed in 236 (89%) of the women who presented, while 30 (11%) were not treated: 15 (6%) due to gas outage, six (2%) due to pregnancy, five (2%) due to concern for cervical cancer, and four (2%) due to an unknown or other reason. After adjusting for other factors in the multivariable analysis, higher education level and missing work to come to the CHC were associated with presenting for treatment. Variables that were associated with increased likelihood of timely presentation were missing work to come to the CHC, absence of depressive symptoms, told by someone important to come to the CHC, and shorter distance to the treatment site.

**Funding:** Research reported in this publication was supported by the National Cancer Institute (https://www.cancer.gov/) of the National Institutes of Health under award number R01CA188428, received by author MH. Additional support was received by author CP from the Charles B. Hammond Research Fund (https://obgyn.duke.edu/research/charles-b-hammond-md-research-fund), Duke University School of Medicine, Durham, NC. About 95% of funding for this project was from the NIH grant and 5% from the Hammond Fund. The content of this publication is solely the responsibility of the authors and does not necessarily represent the official views of the National Institutes of Health. The funders had no role in study design, data collection and analysis, decision to publish, or preparation of the manuscript.

**Competing interests:** The authors have declared that no competing interests exist.

## Conclusion

The majority of hrHPV+ women who did not get treated were lost at the stage of decision-making or accessing treatment, with a small number encountering barriers at the treatment sites. Patient education and financial support are potential areas for intervention to increase rates of hrHPV+ women seeking treatment.

## Introduction

Cervical cancer is the fourth most common cancer in women worldwide and the most common cancer among women in East Africa. This region has the highest incidence of and mortality from cervical cancer in the world; in 2018, age-adjusted mortality was estimated at 16 in 100,000 in East Africa compared to 1 per 100,000 in North America [1]. In Kenya, screening coverage is only 3.5% [2]. Most low-resource countries cannot provide the cytology-based screening that has dramatically reduced cervical cancer mortality in wealthy countries. Several alternative screening technologies are recommended for low-resource settings, and the most effective of these at reducing cervical cancer mortality is testing for high-risk human papillomavirus (hrHPV) [3, 4]. To be effective, hrHPV testing must be part of a cervical cancer prevention cascade, including education, screening, communication of results, and linkage to treatment. The effectiveness of hrHPV testing is reduced if there are high rates of attrition between screening and treatment.

As hrHPV testing is relatively new in low- and middle-income countries such as Kenya, there is limited data on factors that contribute to women's successful completion of a cervical cancer prevention cascade. Barriers may arise at the steps of women deciding to get treated, women navigating the treatment process, and the system providing treatment. Research by Geng et al on loss to follow-up among HIV patients in East Africa has identified structural barriers–e.g. lack of transportation or money, work responsibilities, and childcare responsibilities–and psychosocial barriers–e.g. stigma–as contributing factors [5, 6]. The aim of this study was to identify patient factors associated with whether women who screened hrHPV+ presented for treatment overall, as well as within 30 days of receiving results ("timely" presentation). A number of factors associated with presenting for treatment and presenting for timely treatment were identified.

## Materials and methods

This prospective cohort study was nested within a two-phase cluster-randomized trial comparing implementation strategies for cervical cancer prevention in Migori County in western Kenya (registered at ClinicalTrials.gov, identifier NCT02124252, https://clinicaltrials.gov/ct2/show/NCT02124252?term=NCT02124252&rank=1; protocol available at https://dx.doi.org/10.17504/protocols.io.6s5heg6) [7]. The screening protocol was based on recommendations by the World Health Organization, and the implementation strategies were informed by previous work in the region [8]. It was the first protocol in Kenya to incorporate hrHPV testing as part of screening through government health facilities. In Phase 1, testing was offered via self-collection at either community health campaigns (CHCs) or health facilities, and all hrHPV+ women were referred to the county hospital for treatment with cryotherapy. Less than 50% of women successfully accessed treatment with this standard referral process. Following Phase 1, the study team worked with key stakeholders in the community and government to develop

a strategy for "enhanced linkage to treatment," which was tested in Phase 2. Components of the enhanced linkage strategy included an increased number of decentralized treatment sites and text message treatment reminders. The current study examined loss to follow-up within the enhanced linkage strategy.

CHCs were conducted sequentially in six rural communities in Migori County, with each CHC offering screening for hrHPV with self-collected specimens for two weeks between February and October 2018. In the weeks prior to the CHCs, study staff met with community leaders and used posters, leaflet, and radio advertising to describe the dates and activities of the CHCs. In order to reach the entire community, each campaign moved to multiple sites over its two-week period, with approximately four days at each site. Given this recruitment strategy, the women who registered at the CHCs and enrolled in the study can be considered representative of the women in the six target communities. At the campaigns, women self-tested for hrHPV after receiving education about HPV and cervical cancer. Self-collected specimens are acceptable to women and accurate in detecting hrHPV when compared to clinician-collected specimens [9]. The hrHPV test used was Aptima™ (Hologic/Genprobe Inc.), which can detect the RNA of 14 hrHPV types, including 16 and 18. Women were notified of their hrHPV results and given instructions for follow-up by text message, phone call, or home visit according to their preference.

HrHPV+ women were referred for treatment at one of four government health facilities based on proximity to their community. Treatment was offered for each community starting two weeks after its CHC. As data collection concluded on February 14, 2019, the treatment periods varied in length from 51 weeks for the first community to 21 weeks for the last community. Unless contraindicated by cervical exam, pregnancy, or menses, women were treated by a clinical officer or nurse with cryotherapy, an effective, low-cost treatment modality well-suited to low-resource settings [10]. Women with cervical lesions not amenable to cryotherapy or suspicious for cancer were referred to a gynecologist at Migori County Referral Hospital, in the capital of Migori County.

Data came from two sources: intake questionnaires at the time of screening and treatment questionnaires. At the CHCs, women who provided informed consent completed intake questionnaires administered prior to and after hrHPV screening. Data for all the predictor variables (see below) were collected by the intake questionnaire. Women who presented for treatment and had been consented previously completed a questionnaire prior to treatment, regardless of final eligibility for treatment that day. Questionnaires were verbally administered by research assistants, community health volunteers, nurses, or clinical officers, with data entered directly into tablets using ODK Collect (opendatakit.org).

The study population for the main trial were women in Migori County who were eligible for cervical cancer screening based on the Kenya Ministry of Health's guidelines, i.e. women aged 25–65 years [11]. Pregnant women were excluded. Women were included in this analysis if they consented to participate in Phase 2 of the study, screened for hrHPV at a CHC, tested hrHPV+, and were notified of their result.

The primary outcome was presentation for treatment, regardless of whether it was received. Women were classified as presenting for treatment if they completed a treatment questionnaire or if they called the study team indicating that they had been turned away from the treatment site prior to completion of the questionnaire. The secondary outcome was presenting within 30 days of notification of their hrHPV result ("timely" presentation) versus presenting later than 30 days. This outcome was studied because we hypothesize that delay in treatment leads to greater attrition.

The predictor variables for both the primary and secondary outcomes were those from patient questionnaires that were hypothesized to be associated with presenting for treatment,

as informed by Geng's research [5, 6]. Distance was the geodetic distance (the length of the shortest curve between two points along the surface of a mathematical model of the earth) between the patient's home and the treatment site to which they were referred, calculated from GPS coordinates. The numeric variables of children less than 13 and total children were converted to categorical variables based on the distributions of the primary outcome versus each of these variables.

Data were missing for the variables distance, relationship status, personal cellphone, and would recommend hrHPV self-test to a friend for 26%, 1%, 1%, and 0.2% of participants, respectively. Missing data were imputed using the multivariate normal distribution method of multiple imputation. Fifteen imputed datasets were created, and community was used as an auxiliary variable. The imputed data were included in the bivariate and multivariable analyses. Bivariate associations between the primary and secondary outcomes and each predictor variable were examined, in turn, using binomial logistic regression, to yield unadjusted odds ratios. Age and all predictor variables associated with each outcome with a p-value of <0.10 were included in multivariable logistic regression models for the primary and secondary outcome. These models yielded adjusted odds ratios. In the analysis for the secondary outcome, five participants who presented for treatment were excluded due to unknown date of presentation. For all statistical tests, a two-sided p-value <0.05 was considered significant. All analyses were performed using STATA/SE 15.0.

This study was approved by the Duke Institutional Review Board (protocol # Pro00077442) and the Scientific and Ethics Review Unit of the Kenya Medical Research Institute (protocol # 2918). Participants provided written informed consent at the time of screening and verbal affirmation at each follow-up encounter. For women with lower literacy levels, consent was confirmed with a fingerprint.

## Results

The flow diagram of women included in the study is presented in Fig 1. The rate of hrHPV positivity was 17%, and 92% of hrHPV+ women were successfully notified of their results. A total of 505 women were included in the analysis, of whom 266 (53%) presented for treatment. Of those who presented for treatment, 236 (89%) were treated: 229 (97%) at their first visit and seven (3%) at their second visit. Thirty women (11%) of those who presented were not treated: 15 (6%) due to gas outage, six (2%) due to pregnancy, five (2%) due to concern for cervical cancer, and four (2%) due to an unknown or other reason.

Among study participants, high parity was common (Table 1). The vast majority of women had a primary school education or less. Most were partnered and used cellphones, and slightly more than half worked outside their homes. The majority missed work to attend the CHCs for screening, and only 2% used paid transportation to get there. Six percent of women who did not work outside the home reported missing work to come to the CHC, and 57% of women who worked outside the home reported doing so. The median distance to their treatment site was eight kilometers. While very few reported a diagnosis of depression, 73% reported feeling depressed, down, or hopeless. Most participants were encouraged by a partner, family member, or someone else important to them to come to the CHC, with very few reporting being advised against going. The vast majority said they would recommend hrHPV testing to a friend and would definitely present for treatment if found to be hrHPV+.

The following variables were statistically significantly associated with presenting for treatment: fewer children under age 13, higher education level, missing work to come to the CHC, being told by someone important to come to the CHC, and intention to come for treatment if hrHPV+ (Table 2). Only education level and missing work to come to the CHC remained

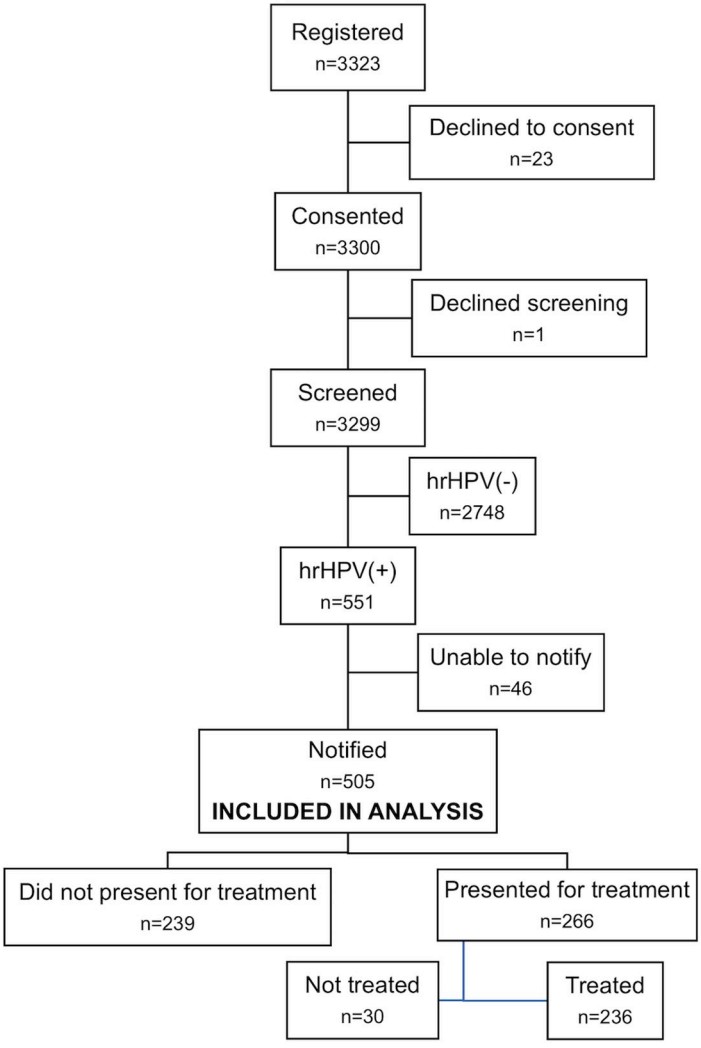

**Fig 1. Flow diagram of study participants, beginning with all women who registered at community health campaigns.** Abbreviations: hrHPV = high-risk human papillomavirus.

statistically significant in the multivariable analysis. The odds of women with at least some secondary education presenting for treatment were 2.38 (95% confidence interval [CI] 1.37–4.13) times the odds of women with a primary school education or less presenting for treatment. Women who missed work to come to the CHC had 1.59 (95% CI 1.08–2.33) times the odds of presenting for treatment than women who did not miss work to come to the CHC.

Similar variables were significantly associated with timely presentation on bivariate and multivariate analysis, including missed work to come to the CHC, frequency of depressive symptoms, told by someone important to come to the CHC, and distance to the treatment site (Table 3). Women who missed work to attend the CHC were more likely to present for earlier treatment than those who did not (OR 2.31, 95% CI 1.13–4.73). Women who reported depressive symptoms some days had 0.43 (95% CI 0.18–0.99) times the odds of presenting for treatment within 30 days as women who reported never having these feelings. There was no significant difference between women who reportedly had these sentiments most or almost every day and those who never experienced them (OR 0.64, 95% CI 0.24–1.72). Women who

**Table 1. Baseline characteristics of hrHPV positive women screened through a cervical cancer prevention program in Migori County, Kenya (N = 505).**

| Patient factor | Category | Median (interquartile range) or frequency (%) |
|---|---|---|
| Age (years) | - - | 33 (27–42) |
| Community of residence | Lwanda | 60 (12%) |
| | Olasi | 114 (23%) |
| | Kituka | 115 (23%) |
| | Kabuto | 86 (17%) |
| | Osingo | 65 (13%) |
| | Ogwedhi | 65 (13%) |
| Total children | 0–2 | 134 (27%) |
| | 3–4 | 166 (33%) |
| | 5+ | 205 (41%) |
| Children under age 13 | 0–2 | 306 (61%) |
| | 3–4 | 171 (34%) |
| | 5+ | 28 (6%) |
| Education level | Primary school or less | 428 (85%) |
| | At least some secondary | 77 (15%) |
| Relationship status | Not partnered | 132 (27%) |
| | Partnered | 366 (73%) |
| Uses a cellphone | No | 87 (17%) |
| | Yes | 418 (83%) |
| Has a personal cellphone | No | 135 (27%) |
| | Yes | 365 (73%) |
| Works outside the home | No | 214 (42%) |
| | Yes | 291 (58%) |
| Missed work to come to the CHC for screening | No | 328 (65%) |
| | Yes | 177 (35%) |
| Used paid transportation to get to the CHC | No | 494 (98%) |
| | Yes | 11 (2%) |
| Frequency of depressive symptoms | Never | 140 (28%) |
| | Some days | 227 (45%) |
| | Most or almost every day | 138 (27%) |
| Ever diagnosed with depression | No | 467 (92%) |
| | Yes | 38 (8%) |
| Told by partner, family member, or someone else important to come to the CHC | No | 189 (37%) |
| | Yes | 316 (63%) |
| Told by partner, family member, or someone else important _not_ to come to the CHC | No | 494 (98%) |
| | Yes | 11 (2%) |
| Would recommend hrHPV self-test to a friend | No | 4 (1%) |
| | Yes | 500 (99%) |
| Reported likelihood (at time of CHC) of seeking treatment if hrHPV+ | Definitely won't | 38 (8%) |
| | Probably will | 35 (7%) |
| | Definitely will | 432 (86%) |
| Distance to treatment site (kilometers) | - - | 8 (5–12) |

Abbreviations: CHC = community health campaign, hrHPV = high-risk human papillomavirus

**Table 2. Odds ratios of presenting for treatment.**

| Patient factor | Category | Did not present for treatment* n = 239 (47%) | Presented for treatment* n = 266 (53%) | Unadjusted odds ratio[†] | 95% CI of unadjusted odds ratio | Adjusted odds ratio[†] | 95% CI of adjusted odds ratio |
|---|---|---|---|---|---|---|---|
| Age (years) | - - | 33 (27–41) | 33 (28–42) | 1.01 | 0.99–1.02 | 1.01 | 0.99–1.03 |
| Community of residence | Lwanda | 27 (45%) | 33 (55%) | 1.00[‡] | | | |
| | Olasi | 57 (50%) | 57 (50%) | 0.82 | 0.44–1.53 | | |
| | Kituka | 53 (46%) | 62 (54%) | 0.96 | 0.51–1.79 | | |
| | Kabuto | 44 (51%) | 42 (49%) | 0.78 | 0.40–1.51 | | |
| | Osingo | 25 (38%) | 40 (62%) | 1.31 | 0.64–2.67 | | |
| | Ogwedhi | 33 (51%) | 32 (49%) | 0.79 | 0.39–1.60 | | |
| Total children | 0–2 | 66 (49%) | 68 (51%) | 1.00[‡] | | | |
| | 3–4 | 85 (51%) | 81 (49%) | 0.92 | 0.59–1.46 | | |
| | 5+ | 88 (43%) | 117 (57%) | 1.29 | 0.83–2.00 | | |
| Children under age 13 | 0–2 | 133 (43%) | 173 (57%) | 1.00[‡] | | 1.00[‡] | |
| | 3–4 | 91 (53%) | 80 (47%) | 0.68 | 0.46–0.98[§] | 0.83 | 0.55–1.25 |
| | 5+ | 15 (54%) | 13 (46%) | 0.67 | 0.31–1.45 | 0.70 | 0.32–1.56 |
| Education level | Primary school or less | 216 (50%) | 212 (50%) | 1.00[‡] | | 1.00[‡] | |
| | At least some secondary | 23 (30%) | 54 (70%) | 2.39 | 1.42–4.04[§] | 2.38 | 1.37–4.13[§] |
| Relationship status | Not partnered | 61 (46%) | 71 (54%) | 1.00[‡] | | | |
| | Partnered | 175 (48%) | 191 (52%) | 0.95 | 0.63–1.41 | | |
| Uses a cellphone | No | 40 (46%) | 47 (54%) | 1.00[‡] | | | |
| | Yes | 199 (48%) | 219 (52%) | 0.94 | 0.59–1.49 | | |
| Has a personal cellphone | No | 67 (50%) | 68 (50%) | 1.00[‡] | | | |
| | Yes | 168 (46%) | 197 (54%) | 1.16 | 0.78–1.72 | | |
| Works outside the home | No | 108 (50%) | 106 (50%) | 1.00[‡] | | | |
| | Yes | 131 (45%) | 160 (55%) | 1.24 | 0.87–1.77 | | |
| Missed work to come to the CHC for screening | No | 170 (52%) | 158 (48%) | 1.00[‡] | | 1.00[‡] | |
| | Yes | 69 (39%) | 108 (61%) | 1.68 | 1.16–2.44[§] | 1.59 | 1.08–2.33[§] |
| Used paid transportation to get to the CHC | No | 235 (48%) | 259 (52%) | 1.00[‡] | | | |
| | Yes | 4 (36%) | 7 (64%) | 1.59 | 0.46–5.49 | | |
| Frequency of depressive symptoms | Never | 62 (44%) | 78 (56%) | 1.00[‡] | | | |
| | Some days | 107 (47%) | 120 (53%) | 0.89 | 0.58–1.36 | | |
| | Most or almost every day | 70 (51%) | 68 (49%) | 0.77 | 0.48–1.24 | | |
| Ever diagnosed with depression | No | 223 (48%) | 244 (52%) | 1.00[‡] | | | |
| | Yes | 16 (42%) | 22 (58%) | 1.26 | 0.64–2.45 | | |
| Told by partner, family member, or someone else important to come to the CHC | No | 101 (53%) | 88 (47%) | 1.00[‡] | | 1.00[‡] | |
| | Yes | 138 (44%) | 178 (56%) | 1.48 | 1.03–2.13[§] | 1.30 | 0.89–1.89 |
| Told by partner, family member, or someone else important _not_ to come to the CHC | No | 232 (47%) | 262 (53%) | 1.00[‡] | | | |
| | Yes | 7 (64%) | 4 (36%) | 0.51 | 0.15–1.75 | | |
| Would recommend hrHPV self-test to a friend | No | 2 (50%) | 2 (50%) | 1.00[‡] | | | |
| | Yes | 236 (47%) | 264 (53%) | 1.10 | 0.15–7.86 | | |
| Reported likelihood (at time of CHC) of seeking treatment if hrHPV+ | Definitely won't | 13 (34%) | 25 (66%) | 1.00[‡] | | 1.00[‡] | |
| | Probably will | 22 (63%) | 13 (37%) | 0.31 | 0.12–0.80[§] | 0.40 | 0.15–1.06 |
| | Definitely will | 204 (47%) | 228 (53%) | 0.58 | 0.29–1.17 | 0.61 | 0.30–1.25 |

*(Continued)*

**Table 2.** (Continued)

| Patient factor | Category | Did not present for treatment* n = 239 (47%) | Presented for treatment* n = 266 (53%) | Unadjusted odds ratio† | 95% CI of unadjusted odds ratio | Adjusted odds ratio† | 95% CI of adjusted odds ratio |
|---|---|---|---|---|---|---|---|
| Distance to treatment site (kilometers) | - - | 8 (5–12) | 8 (5–13) | 1.01 | 0.97–1.05 | | |

\* Data expressed as median (interquartile range) or frequency (%)

† With missing data imputed by multiple imputation

‡ Baseline category

§ Differs significantly from 1 at the 95% confidence level

Abbreviations: 95% CI = 95% confidence interval, CHC = community health campaign, hrHPV = high-risk human papillomavirus

were encouraged to come to the CHC by someone important to them were less likely to present for timely treatment than women who did not report this social influence (OR 0.35, 95% CI 0.16–0.77). Finally, for each additional kilometer of distance to the treatment site, the odds of presenting within 30 days decreased by a factor of 0.91 (95% CI 0.85–0.97).

## Discussion

The success of a cervical cancer prevention program depends not just on its basis in effective screening and treatment techniques, but also its understanding of the target population to ensure uptake of services across the entire prevention cascade. In our study, the majority of hrHPV+ women who did not get treated were lost at the stage of decision-making or accessing treatment, with only a small number turned away due to logistical or personnel issues at treatment sites. We found that in a program designed to enhance linkage to treatment with input from key community stakeholders, there were still a number of factors that remained barriers to women presenting for treatment.

Our study identified women with low education level, endorsing depressive symptoms, and living farther from the treatment sites as less likely to present for treatment at all or within 30 days. Women were more likely to present for treatment if they missed work to come for screening, possibly because the ability to miss work is a proxy for socioeconomic status and ability to travel outside their home or village. Surprisingly, women who were encouraged by someone important to them to come to the CHC for screening were less likely to present for timely treatment than those who did not report such encouragement. There are several possible explanations for this counterintuitive finding: there were small numbers of participants in some of the involved categories; women who were encouraged to come to the CHC may have been less personally motivated to pursue screening and treatment; and women who endorsed social support in coming to the CHC may have received less counseling from study providers about the importance of presenting for treatment. On the other hand, women who were encouraged to come to the CHC were *more* likely in the bivariate analysis to present for treatment at all; while this result did not remain significant in the multivariate analysis, we can conclude that any associations between encouragement to come to the CHC and presentation for treatment are ambiguous, and conclusions should be limited by the observational nature of the study and sample size.

While this study adds to the understanding of personal and logistical factors that influence women's ability to get treated for a positive hrHPV result, there are some limitations. The number of women who presented to the treatment sites but did not receive treatment was likely underestimated since this quantification relied on self-report among women calling the

**Table 3. Odds ratios of presenting for treatment within versus later than 30 days ("timely" presentation).**

| Patient factor | Category | Presented within 30 days* n = 205 (79%) | Presented later than 30 days* n = 56 (21%) | Unadjusted odds ratio[†] | 95% CI of unadjusted odds ratio | Adjusted odds ratio[†] | 95% CI of adjusted odds ratio |
|---|---|---|---|---|---|---|---|
| Age (years) | - - | 33 (28–41) | 33 (27–43) | 1.00 | 0.98–1.03 | 1.02 | 0.99–1.05 |
| Community of residence | Lwanda | 27 (82%) | 6 (18%) | 1.00[‡] | | | |
| | Olasi | 39 (70%) | 17 (30%) | 0.51 | 0.18–1.46 | | |
| | Kituka | 44 (72%) | 17 (28%) | 0.58 | 0.20–1.64 | | |
| | Kabuto | 36 (86%) | 6 (14%) | 1.33 | 0.39–4.59 | | |
| | Osingo | 38 (97%) | 1 (3%) | 8.44 | 0.96–74.23 | | |
| | Ogwedhi | 21 (70%) | 9 (30%) | 0.52 | 0.16–1.69 | | |
| Total children | 0–2 | 57 (86%) | 9 (14%) | 1.00[‡] | | | |
| | 3–4 | 61 (76%) | 19 (24%) | 0.51 | 0.21–1.21 | | |
| | 5+ | 87 (76%) | 28 (24%) | 0.49 | 0.22–1.12 | | |
| Children under age 13 | 0–2 | 139 (82%) | 30 (18%) | 1.00[‡] | | | |
| | 3–4 | 57 (72%) | 22 (28%) | 0.56 | 0.30–1.05 | | |
| | 5+ | 9 (69%) | 4 (31%) | 0.49 | 0.14–1.68 | | |
| Education level | Primary school or less | 164 (78%) | 45 (22%) | 1.00[‡] | | | |
| | At least some secondary | 41 (79%) | 11 (21%) | 1.02 | 0.49–2.15 | | |
| Relationship status | Not partnered | 51 (74%) | 18 (26%) | 1.00[‡] | | | |
| | Partnered | 152 (80%) | 37 (20%) | 1.46 | 0.77–2.79 | | |
| Uses a cellphone | No | 39 (85%) | 7 (15%) | 1.00[‡] | | | |
| | Yes | 166 (77%) | 49 (23%) | 0.61 | 0.26–1.44 | | |
| Has a personal cellphone | No | 55 (83%) | 11 (17%) | 1.00[‡] | | | |
| | Yes | 149 (77%) | 45 (23%) | 0.67 | 0.32–1.38 | | |
| Works outside the home | No | 77 (75%) | 25 (25%) | 1.00[‡] | | | |
| | Yes | 128 (81%) | 31 (20%) | 1.34 | 0.74–2.44 | | |
| Missed work to come to the CHC for screening | No | 113 (73%) | 41 (27%) | 1.00[‡] | | 1.00[‡] | |
| | Yes | 92 (86%) | 15 (14%) | 2.23 | 1.16–4.27[§] | 2.31 | 1.13–4.73[§] |
| Used paid transportation to get to the CHC | No | 201 (79%) | 53 (21%) | 1.00[‡] | | | |
| | Yes | 4 (57%) | 3 (43%) | 0.35 | 0.08–1.62 | | |
| Frequency of depressive symptoms | Never | 69 (88%) | 9 (12%) | 1.00[‡] | | 1.00[‡] | |
| | Some days | 84 (72%) | 33 (28%) | 0.33 | 0.15–0.74[§] | 0.43 | 0.18–0.99[§] |
| | Most or almost every day | 52 (79%) | 56 (21%) | 0.48 | 0.19–1.21 | 0.64 | 0.24–1.72 |
| Ever diagnosed with depression | No | 189 (79%) | 50 (21%) | 1.00[‡] | | | |
| | Yes | 16 (73%) | 6 (27%) | 0.71 | 0.26–1.90 | | |
| Told by partner, family member, or someone else important to come to the CHC | No | 76 (88%) | 10 (12%) | 1.00[‡] | | 1.00[‡] | |
| | Yes | 129 (74%) | 46 (26%) | 0.37 | 0.18–0.77[§] | 0.35 | 0.16–0.77[§] |
| Told by partner, family member, or someone else important _not_ to come to the CHC | No | 201 (78%) | 56 (22%) | 1.00[‡] | | | |
| | Yes | 4 (100%) | 0 (0%) | 1 | 1–1 | | |
| Would recommend hrHPV self-test to a friend | No | 1 (50%) | 1 (50%) | 1.00[‡] | | | |
| | Yes | 204 (79%) | 55 (21%) | 3.71 | 0.23–60.25 | | |
| Reported likelihood (at time of CHC) of seeking treatment if hrHPV+ | Definitely won't | 17 (68%) | 8 (32%) | 1.00[‡] | | | |
| | Probably will | 12 (92%) | 1 (8%) | 5.65 | 0.62–51.29 | | |
| | Definitely will | 176 (79%) | 47 (21%) | 1.76 | 0.72–4.33 | | |

(*Continued*)

**Table 3.** (Continued)

| Patient factor | Category | Presented within 30 days* n = 205 (79%) | Presented later than 30 days* n = 56 (21%) | Unadjusted odds ratio† | 95% CI of unadjusted odds ratio | Adjusted odds ratio† | 95% CI of adjusted odds ratio |
|---|---|---|---|---|---|---|---|
| Distance to treatment site (kilometers) | - - | 8 (5–11) | 12 (6–16) | 0.92 | 0.87–0.98§ | 0.91 | 0.85–0.97§ |

* Data expressed as median (interquartile range) or frequency (%)

† With missing data imputed by multiple imputation

‡ Baseline category

§ Differs significantly from 1 at the 95% confidence level

Abbreviations: 95% CI = 95% confidence interval, CHC = community health campaign, hrHPV = high-risk human papillomavirus

study staff. Women who left before being seen because they had waited too long or because there was not a provider were unlikely to have completed a questionnaire. We did not have a good proxy for socioeconomic status, with missing work to come to the CHC probably being the closest. Another limitation was the occurrence of missing data, particularly for distance to the treatment sites; however, multiple imputation was used to address this limitation. In addition, the distance reported was geodetic distance rather than distance by road, with the former method also being used by Geng et al [5].

This study identified a number of potential areas for intervention to increase treatment rates among hrHPV+ women. Low education level and poor understanding of hrHPV and cervical cancer were identified as barriers to presenting for treatment. Knowledge of cervical cancer screening can be improved with a brief educational intervention [12], like what was incorporated into the CHCs in this study; augmenting this intervention might further increase treatment rates. Although treatment was decentralized and patient navigators were employed in the enhanced linkage strategy, women still faced significant financial and logistical transportation barriers. Reimbursing patients for transportation costs or providing mobile treatment might improve access to and should be studied in ways that facilitate sustainable implementation. Furthermore, increasing outreach to women identified as at-risk for loss to follow-up in this study, e.g. those with limited education or reporting depressive symptoms, could help increase treatment rates. These barriers could be addressed to some extent by greater involvement of community health volunteers to educate and motivate hrHPV+ women to get treated, help them access treatment, and provide support at treatment sites.

## Supporting information

**S1 Dataset. De-identified dataset.**
(XLS)

## Acknowledgments

The authors thank Moreen Njoroge of Duke University for assisting with data acquisition.

## Author Contributions

**Conceptualization:** Charlotte M. Page, Megan J. Huchko.

**Data curation:** Saduma Ibrahim.

**Formal analysis:** Charlotte M. Page.

**Methodology:** Lawrence P. Park.

**Supervision:** Megan J. Huchko.

**Writing – original draft:** Charlotte M. Page.

**Writing – review & editing:** Charlotte M. Page, Saduma Ibrahim, Lawrence P. Park, Megan J. Huchko.

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
