## [Decision Letter · Decision Letter 0]

1 Aug 2019

PONE-D-19-17741

Patient factors affecting successful linkage to treatment in a cervical cancer prevention program in Kenya: a prospective cohort study

PLOS ONE

Dear Dr. Page,

Thank you for submitting your manuscript to PLOS ONE. After careful consideration, we feel that it has merit but does not fully meet PLOS ONE’s publication criteria as it currently stands. Therefore, we invite you to submit a revised version of the manuscript that addresses the points raised during the review process.

We would appreciate receiving your revised manuscript by Sep 15 2019 11:59PM. To enhance the reproducibility of your results, we recommend that if applicable you deposit your laboratory protocols in protocols.io, where a protocol can be assigned its own identifier (DOI) such that it can be cited independently in the future. For instructions see: http://journals.plos.org/plosone/s/submission-guidelines#loc-laboratory-protocols

We look forward to receiving your revised manuscript.

Kind regards,

Violet Naanyu

Academic Editor

PLOS ONE

Journal Requirements:

2. In your Methods section, please provide additional information about the participant recruitment method, including a statement as to whether your sample can be considered representative of a larger population, and a description of how participants were recruited.

Reviewers' comments:

Reviewer's Responses to Questions

**Comments to the Author**

1. Is the manuscript technically sound, and do the data support the conclusions?

Reviewer #1: Yes

Reviewer #2: Partly

2. Has the statistical analysis been performed appropriately and rigorously? 

Reviewer #1: Yes

Reviewer #2: No

3. Have the authors made all data underlying the findings in their manuscript fully available?

Reviewer #1: No

Reviewer #2: Yes

4. Is the manuscript presented in an intelligible fashion and written in standard English?

Reviewer #1: Yes

Reviewer #2: Yes

5. Review Comments to the Author

Reviewer #1: Line 102- Women with cervical lesions not amenable to cryotherapy or

103 suspicious for cancer were referred to a gynecologist. What was the distance women had to travel to see a gynecologist and what was the waiting time?

Line 112- The study population for the main trial were women in Migori County who were eligible

113 for cervical cancer screening based on the Kenya Ministry of Health’s guidelines: 25-65 years old with an intact uterus and cervix. – add reference

Reviewer #2: Reviewer Comments

“Patient factors affecting successful linkage to treatment in a cervical cancer prevention program in Kenya: a prospective cohort study”

General comment

• The manuscript is well written however there are some comments in each of the section that could improve the manuscript.

Abstract

• In the results section provide direction of association for the significant variables whether positive or negative

Introduction

More background need to be provided on the linkage rate after screening in the region and in the whole country in general. This is not clear before assessing factors what is the burden.

Methodology

Were the participants counseled prior or after testing?

In the methods it is stated that this current manuscript focuses on the group that was exposed to enhanced linkage strategy that included transport facilitation for a limited number of participants. How many among the one studied had facilitated transportation? How was this handled in the analysis.

Results

• The variable Missed work to come to the CHC for screening was this for women who were employed or for all women

• How were the variables “Told by partner, family member, or someone else important to come to the CHC” and “Told by partner, family member, or someone else important not to come to the CHC” phrased since one is the negative of the other.

• “Women with a college education or beyond were not significantly more likely to present for treatment than women in the least educated category (OR 2.17, 95% CI 0.65-7.26), with a wide confidence interval in the setting of very few women having a college education or more”. Why not consider collapsing the groups?

• Was multicollinearity before the adjusted odds ratio were obtained.

Data Availability

Indicated that the data is available but no link to the data

6. PLOS authors have the option to publish the peer review history of their article (what does this mean?). If published, this will include your full peer review and any attached files.

Reviewer #1: Yes: Elkanah Omenge Orang'o

Reviewer #2: No

---

## [Author Response · Author response to Decision Letter 0]

9 Aug 2019

Response: We thank the editor and reviewers for your careful reading of our manuscript and for your thoughtful comments. We have incorporated your suggestions into the updated version of our manuscript as detailed below. Please note that line numbers in our responses below refer to the tracked version of the revised manuscript.

Responses to comments from the editor:

http://www.journals.plos.org/plosone/s/file?id=wjVg/PLOSOne_formatting_sample_main_body.pdf andhttp://www.journals.plos.org/plosone/s/file?id=ba62/PLOSOne_formatting_sample_title_authors_affiliations.pdf

Response: We have reviewed PLOS ONE’s style requirements to ensure that our manuscript is compliant. We made changes in the punctuation around the in-text citations. 

2. In your Methods section, please provide additional information about the participant recruitment method, including a statement as to whether your sample can be considered representative of a larger population, and a description of how participants were recruited.

Response: Thank you for this suggestion. We have added the requested information to the Methods section (lines 103-8). 

Response: There are no ethical restrictions to sharing a de-identified dataset. We now include this dataset as Supporting Information file “S2 Dataset.” 

Responses to comments from the reviewers:

Reviewer #1: Line 102- Women with cervical lesions not amenable to cryotherapy or

103 suspicious for cancer were referred to a gynecologist. What was the distance women had to travel to see a gynecologist and what was the waiting time?

Response: These are excellent questions. Women were referred to a gynecologist at Migori County Referral Hospital, which is in the capital of Migori County. We have added this information to our Methods section (lines 122-3). The geodetic distance to this site from the homes of women in the study population ranges from 4 to 40 km, with a median of 29 km. We do not have information on the waiting time; however, the supervising gynecologist was a consultant on the study and worked to facilitate care for all referred women in a timely fashion. 

Line 112- The study population for the main trial were women in Migori County who were eligible

113 for cervical cancer screening based on the Kenya Ministry of Health’s guidelines: 25-65 years old with an intact uterus and cervix. – add reference

Response: We have added a reference and removed the statement about “an intact uterus and cervix” after clarifying against the reference (lines 136-7). 

Reviewer #2: Reviewer Comments

“Patient factors affecting successful linkage to treatment in a cervical cancer prevention program in Kenya: a prospective cohort study”

General comment

• The manuscript is well written however there are some comments in each of the section that could improve the manuscript.

Abstract

• In the results section provide direction of association for the significant variables whether positive or negative

Response: This is a good suggestion. We have updated the abstract accordingly (lines 39-42). 

Introduction

More background need to be provided on the linkage rate after screening in the region and in the whole country in general. This is not clear before assessing factors what is the burden.

Response: Since this is the first protocol in Kenya to incorporate hrHPV testing as part of screening through government health facilities, there is no linkage to treatment rate to which to compare outside of this study. However, in Phase 1 of this study, which employed standard linkage to treatment, less than 50% of women were successfully linked. We have included this data in the Methods section (lines 89-90). 

Methodology

Were the participants counseled prior or after testing?

Response: The participants were counseled prior to testing. We have provided clarification about this counseling in lines 108-9. 

In the methods it is stated that this current manuscript focuses on the group that was exposed to enhanced linkage strategy that included transport facilitation for a limited number of participants. How many among the one studied had facilitated transportation? How was this handled in the analysis.

Response: We clarified with the study coordinator, and there actually was no transport facilitation for women in Phase 2 of the study, which is the phase of interest for this research. We apologize for this error and have removed this statement from the manuscript (line 94). 

Results

• The variable Missed work to come to the CHC for screening was this for women who were employed or for all women

Response: This variable applied to all women. We think there is something about the decision and ability to miss work that separates women who missed work from those who either work inside the home or work outside the home and didn’t miss work. In preparing our response to this question, we realized that the frequencies in Table 1 for this variable were incorrect; they are now accurate. 

• How were the variables “Told by partner, family member, or someone else important to come to the CHC” and “Told by partner, family member, or someone else important not to come to the CHC” phrased since one is the negative of the other.

Response: The questions were worded to participants at the CHCs as follows: “Did your partner, family member, or anyone else important to you tell you to come?” and “Did your partner, family member, or anyone else important to you tell you not to come?” During training and piloting of the surveys, we had the research assistants stress the difference between the two questions.

• “Women with a college education or beyond were not significantly more likely to present for treatment than women in the least educated category (OR 2.17, 95% CI 0.65-7.26), with a wide confidence interval in the setting of very few women having a college education or more”. Why not consider collapsing the groups?

Response: It would certainly be reasonable to combine the groups “at least some secondary” and “college or beyond”; however, we chose to keep them separate to provide more detailed data on the distribution of education levels in the study population. 

• Was multicollinearity before the adjusted odds ratio were obtained.

Response: Yes, we did test for multicollinearity among the variables that we used in the multivariable logistic regression models. The variance inflation factor (VIN) for all variables was <2, indicating that collinearity is not a concern. 

Data Availability

Indicated that the data is available but no link to the data

Response: We now attach the de-identified data as a Supporting Information file.

---

## [Decision Letter · Decision Letter 1]

22 Aug 2019

PONE-D-19-17741R1

Patient factors affecting successful linkage to treatment in a cervical cancer prevention program in Kenya: a prospective cohort study

PLOS ONE

Dear Charlotte M Page,

Thank you for submitting your manuscript to PLOS ONE. After careful consideration, we feel that it has merit but does not fully meet PLOS ONE’s publication criteria as it currently stands. Therefore, we invite you to submit a revised version of the manuscript that addresses the minor points raised during the second review process.

We would appreciate receiving your revised manuscript by August 30, 2019. To enhance the reproducibility of your results, we recommend that if applicable you deposit your laboratory protocols in protocols.io, where a protocol can be assigned its own identifier (DOI) such that it can be cited independently in the future. For instructions see: http://journals.plos.org/plosone/s/submission-guidelines#loc-laboratory-protocols

We look forward to receiving your revised manuscript.

Kind regards,

Violet Naanyu

Academic Editor

PLOS ONE

Additional Editor Comments (if provided):

Dear Authors,

Thank you for a thorough review and editing of the manuscript. The reader gets a full picture of the study activities, associated analyses, and conclusions reached.

Reviewers' comments:

Reviewer's Responses to Questions

**Comments to the Author**

1. If the authors have adequately addressed your comments raised in a previous round of review and you feel that this manuscript is now acceptable for publication, you may indicate that here to bypass the “Comments to the Author” section, enter your conflict of interest statement in the “Confidential to Editor” section, and submit your "Accept" recommendation.

Reviewer #1: All comments have been addressed

Reviewer #2: (No Response)

2. Is the manuscript technically sound, and do the data support the conclusions?

Reviewer #1: Yes

Reviewer #2: Yes

3. Has the statistical analysis been performed appropriately and rigorously? 

Reviewer #1: Yes

Reviewer #2: No

4. Have the authors made all data underlying the findings in their manuscript fully available?

Reviewer #1: Yes

Reviewer #2: Yes

5. Is the manuscript presented in an intelligible fashion and written in standard English?

Reviewer #1: Yes

Reviewer #2: Yes

6. Review Comments to the Author

Reviewer #1: This is a well written paper with interesting findings that may inform similar settings in as far as cervical cancer screening and treatment is concerned.

Reviewer #2: (No Response)

7. PLOS authors have the option to publish the peer review history of their article (what does this mean?). If published, this will include your full peer review and any attached files.

Reviewer #1: Yes: Orang'o Elkanah Omenge

Reviewer #2: No

---

## [Author Response · Author response to Decision Letter 1]

28 Aug 2019

Manuscript: PONE-D-19-17741

Response: We thank the editor and reviewers for your careful reading of our manuscript and for your thoughtful comments. We have incorporated your suggestions into the updated version of our manuscript as detailed below. Please note that line numbers in our responses below refer to the tracked version of the revised manuscript.

Responses to comments from Reviewer 2:

1. The variable Missed work to come to the CHC for screening was this for women who were employed or for all women 

Response: This variable applied to all women. We think there is something about the decision and ability to miss work that separates women who missed work from those who either work inside the home or work outside the home and didn’t miss work. In preparing our response to this question, we realized that the frequencies in Table 1 for this variable were incorrect; they are now accurate. 

Comment: How many among those working outside home missed work to come to CCC. Since the ones at home might not have considered themselves to have missed work. This all depends on how the questions was phrased and from the frequencies the number missing work is lower than the number working outside home.

Response: The questions were phrased as, “Are you currently working outside of the home?” and “Did you have to miss work to come to this campaign?” Six percent of women who do not work outside the home reported missing work to come to the CHC, and 57% of women who work outside the home reported doing so. We have added this data to the text of the manuscript (lines 177-9). The 2x2 table is below. 

 Missed work Did NOT miss work

Work outside the home 165 126

Do NOT work outside the home 12 202

When the analysis was limited to the 291 women who work outside the home, the association between missing work to come to the CHC and presenting for treatment remained significantly significant (2.01, 95% confidence interval 1.26-3.22). 

2. “Women with a college education or beyond were not significantly more likely to present for treatment than women in the least educated category (OR 2.17, 95% CI 0.65-7.26), with a wide confidence interval in the setting of very few women having a college education or more”. Why not consider collapsing the groups? 

Response: It would certainly be reasonable to combine the groups “at least some secondary” and “college or beyond”; however, we chose to keep them separate to provide more detailed data on the distribution of education levels in the study population.

 Comment: Due to the small numbers in the College group why not combine them with the secondary to gain more power. Then we can be able to make the conclusion that education is a factor currently its only between secondary and primary.

Response: We have combined the college group with the secondary education group and updated the statistical analysis to reflect this change.

---

## [Editor Report · Decision Letter 2]

9 Sep 2019

Patient factors affecting successful linkage to treatment in a cervical cancer prevention program in Kenya: a prospective cohort study

PONE-D-19-17741R2

Dear Charlotte Page,

We are pleased to inform you that your manuscript has been judged scientifically suitable for publication and will be formally accepted for publication once it complies with all outstanding technical requirements.

With kind regards,

Violet Naanyu

Academic Editor

PLOS ONE

---

## [Editor Report · Acceptance letter]

11 Sep 2019

PONE-D-19-17741R2 

Patient factors affecting successful linkage to treatment in a cervical cancer prevention program in Kenya: a prospective cohort study 

Dear Dr. Page:

I am pleased to inform you that your manuscript has been deemed suitable for publication in PLOS ONE. Congratulations! Your manuscript is now with our production department. 

With kind regards,

on behalf of

Prof. Violet Naanyu 

Academic Editor

PLOS ONE